# Reactive Glia Inflammatory Signaling Pathways and Epilepsy

**DOI:** 10.3390/ijms21114096

**Published:** 2020-06-08

**Authors:** Pascual Sanz, Maria Adelaida Garcia-Gimeno

**Affiliations:** 1Instituto de Biomedicina de Valencia (CSIC) and Centro de Investigación Biomédica en Red de Enfermedades Raras (CIBERER), Jaime Roig 11, 46010 Valencia, Spain; 2Department of Biotechnology, Escuela Técnica Superior de Ingeniería Agronómica y del Medio Natural (ETSIAMN), Universitat Politècnica de València, 46022 Valencia, Spain; agarcia@ibv.csic.es

**Keywords:** inflammation, signaling pathways, epilepsy, reactive astrocytes, microglia

## Abstract

Neuroinflammation and epilepsy are interconnected. Brain inflammation promotes neuronal hyper-excitability and seizures, and dysregulation in the glia immune-inflammatory function is a common factor that predisposes or contributes to the generation of seizures. At the same time, acute seizures upregulate the production of pro-inflammatory cytokines in microglia and astrocytes, triggering a downstream cascade of inflammatory mediators. Therefore, epileptic seizures and inflammatory mediators form a vicious positive feedback loop, reinforcing each other. In this work, we have reviewed the main glial signaling pathways involved in neuroinflammation, how they are affected in epileptic conditions, and the therapeutic opportunities they offer to prevent these disorders.

## 1. Introduction

Epilepsy is a neurological disorder characterized by a predisposition to generate recurrent epileptic seizures and the associated cognitive, psychological, and social consequences [1]. Epilepsy affects 1% of the total world population (around 65 million people worldwide) and it is caused by acquired insults in the brain (e.g., after stroke or traumatic brain injury), infectious diseases, autoimmune diseases, and genetic mutations [1,2]. The first-line treatment for epilepsy are anti-seizure drugs (ASDs). The development of ASDs was based on the neuron-centric hypothesis that an imbalance of excitatory and inhibitory currents was largely responsible for epileptic seizures [3]. However, despite the availability of many ASDs, approximately one-third of patients fail to achieve seizure control or soon become resistant to the effects of the ASDs [1,3]. Consequently, there is a critical need for the development of innovative anti-epileptogenic treatment strategies to ameliorate the progression or/and limit the detrimental consequences of the disease [1,4].

Recently, a critical role for glia (astrocytes, microglia, and oligodendrocytes) has been reported in the development of different neurodegenerative diseases [5]. Therefore, glial cells are no longer considered just bystanders of brain function but they are now considered critical players in brain pathophysiology. In fact, an astrocytic basis for epilepsy has been proposed and results obtained both in animal models and also in human samples indicate that astrocyte dysfunction can participate in hyper-excitation, neurotoxicity, and seizure spreading, on top of their established neurogenic functions [5]. Perhaps, this is the reason why the European Commission of the International League Against Epilepsy (ILAE) recognized the work on the role that glia and inflammation may have on the development of seizures and epileptogenesis as the highest research priority and encouraged the identification of glial targets as a basis for the development of more specific anti-epileptogenic drugs [6].

Glial cells are the earlier sensors of brain abnormalities. Upon a brain insult, astrocytes and microglia become reactive and this precedes the appearance of the neuropathological symptoms [7]. After activation, both astrocytes and microglia secrete pro-inflammatory mediators to initially protect, adapt, and return the central nervous system (CNS) to its regular function. However, if the insult is maintained, the release of pro-inflammatory mediators is harmful, since the persistent activation of inflammatory pathways exacerbates the neuropathological processes [8,9,10].

It is becoming clear that brain inflammation promotes neuronal hyper-excitability and seizures and that dysregulation in the glia immune-inflammatory function is a common factor that predisposes or contributes to the generation of seizures. At the same time, acute seizures upregulate the production of pro-inflammatory cytokines in microglia and astrocytes, triggering a downstream cascade of inflammatory mediators. Therefore, epileptic seizures and inflammatory mediators form a vicious positive feedback loop, reinforcing each other [11]. For this reason, it has been recently proposed that targeting inflammation with specific anti-inflammatory drugs may be beneficial in the treatment of refractory epilepsies [11]. However, before selecting any type of drug, a deep knowledge of the main affected inflammatory pathways has to be gained in each particular type of epilepsy.

In this work, we review the main glial signaling pathways involved in neuroinflammation, how they are affected in epileptic conditions, and the therapeutic opportunities they offer to prevent these disorders.

## 2. Glial Inflammatory Pathways and Epilepsy

As in the case of other cells involved in cellular inflammation (i.e., leukocytes, dendritic cells, etc.), glial cells may react to exogenous (pathogen-associated molecular patterns, PAMPs) or endogenous inflammatory inducers (damage-associated molecular patterns (DAMPs), such as ATP, advance glycation end products (AGEs), high mobility group box 1 (HMGB1; a non-histone chromatin-binding protein that is released upon inflammatory conditions), and S100β (a Ca++ binding protein)) [12,13]. These inducers interact with specific pattern recognition receptors (PRRs) located either in the glial membrane (e.g., Toll-like receptors (TLRs; IL-1R1 (interleukin1 receptor1), RAGEs (receptors of AGEs)) or in the cytosol (e.g., nucleotide-binding oligomerization domain (NOD) and leucine-rich repeat receptors (NLRs), and absent in melanoma 2 (AIM2)-like receptors (ALRs)) [12,14]. These interactions trigger specific signaling cascades that stimulate different pro-inflammatory mediators, e.g., nuclear factor kappa-light-chain-enhancer of activated B cells (NF-kB), interferon regulatory factor 3 (IRF3) family of transcription factors, mitogen-activated protein kinase (MAPK) signaling pathways (p38, extracellular signal-regulated kinase (ERK), Jun N-terminal kinase (JNK)), leading to the expression of pro-inflammatory agents (e.g., pro-interleukin1-β (pro-IL-1β), interleukin-6 (IL-6), tumor necrosis factor α (TNFα), chemokines, typeI-interferons (typeI-IFNs), etc.) [12,13,14,15] (see Figure 1, Figure 2, Figure 3, Figure 4 and Figure 5 below). In general, these signaling cascades end with the death of the cell by pyroptosis (from the Greek pyros: fire, inflammation; and ptosis: death), which is highly pro-inflammatory since it is accompanied by the release of some DAMPs including interleukin1-α (IL-1α) and HMGB1 [15,16]. When the inducer that triggers the inflammatory reaction does not have a microbial nature, then the reaction is known as sterile inflammation [14]. 

Some of the main glial signaling pathways involved in inflammation are discussed below.

### 2.1. Toll-Like Receptor Signaling Pathway

Toll-like receptors (TLRs) are dimeric membrane proteins located either in the plasma membrane or in endosomal membranes which respond to specific inducers. The main TLRs present in glial cells are the plasma membrane located TLR2 (which mainly responds to β-glucans), TLR4 (which responds to bacterial lipopolysaccharides (LPS) and to high mobility group 1 (HMGB1) protein), IL-1R1 (which respond to IL-1β) and RAGE (receptor of advanced glycation end products (AGEs), which also responds to HMGB1, among other inducers), as well as the endosomal TLR3 (which responds to the presence of polynucleotides, e.g., poly I:C) [10,12,17,18,19]. The activation of these receptors by their specific inducers triggers several signaling pathways:

#### 2.1.1. MyD88-Dependent Pathway

In brief, in the cases of TLR2, TLR4, IL-1R1, and RAGE, activation of these receptors results in the binding of the MyD88 adaptor to the cytosolic domain of the receptors. This allows the binding of the IL-1 receptor-associated kinase (IRAK) family of Ser/Thr protein kinases (especially IRAK1/2/4) forming the myddosome. Then the E3-ubiquitin ligase TRAF6 (TNFR (tumor necrosis factor receptor)-associated factor 6) binds to the myddosome and activates the TAK1 (transforming growth factor b-activating kinase)/TAB1-TAB2-TAB3 (TAK1/2/3 binding proteins) kinase complex resulting, on the one hand, in the activation of the IKKs-NEMO (IKKα-IKKβ-IKKγ (inhibitor of the kappa light polypeptide gene enhancer in B-cells kinase)/NEMO (NF-kB essential modulator)) kinase complex, leading to the phosphorylation of IkBs and the release of the p65-p50 NF-kB (nuclear factor kappa-light-chain-enhancer of activated B cells) proteins. On the other hand, the TAK1/TABs(1,2,3) kinase complex activates several MAPK pathways (ERK, JNK, and p38) resulting in the activation of the AP-1 (activator protein 1, formed by c-fos/c-Jun) transcription factor [20]. Both NF-kB and AP-1 will lead to the activation of the expression of pro-inflammatory mediators (e.g., pro-IL-1β, IL-6, TNFα, chemokines, etc.) [12,19] (Figure 1).

#### 2.1.2. MyD88-Independent Pathway

In the case of endosomal TLR3 activation, its cytosolic domain interacts with the TRIF (TIR (Toll Interleukin receptor) domain-containing adaptor protein-inducing interferon β)-TRAM (*TRIF*-related adaptor molecule) adaptors, which results in the recruiting of TRAF3 (TNFR-associated factor 3) and the subsequent activation of the TBK-1 [TANK (TRAF-associated NF-kB activator)-binding] kinase, that will activate the IRF3 (interferon regulatory factor 3) transcription factor leading to the expression of typeI-IFNs (interferons) IFNα and IFNβ. In parallel, the TRIF-TRAM complex will activate TRAF6, leading to the activation of the TAK1/TABs(1,2,3) complex described above [12,19,21] (Figure 1).

The final consequence of the sustained activation of the TLR pathways is the death of the cell by pyroptosis (see Section 2.2). In this process, cells produce large amounts of pro-IL-1β, pro-IL-18, and gasdermin D, which are processed to IL-1β, IL-18, and gasdermin-N, the latter affecting plasma membrane stability. Cells release then IL-1β, IL-18, and other intracellular proteins, such as IL-1α and HMGB1, which act as DAMPs on the surrounding cells. [12,19].

It has been described that activation of the TLR4 and IL-1R1 receptors plays a key role in epileptogenesis. In fact, patients suffering from mesial temporal lobe epilepsy (MTLE) present higher levels of TLR4, IL-1R1, IL-1β, and HMGB1 in their hippocampus (reviewed in [17]) (Figure 1, names in blue). Changes in the levels of HMGB1 in the brain are mirrored in blood, so measuring blood levels of HMGB1 in patients can predict with high accuracy the risk of developing epilepsy (reviewed in [22]). In the same way, changes in the levels of IL-1β in blood or cerebrospinal fluid (CSF) in human samples could also be used as biomarkers of the progression of the disease (reviewed in [23]).

In order to prevent the overactivation of the TLRs, several strategies have been developed:

(a) To reduce the IL-1β/IL-1R1 signaling pathway several authors have used anakinra (an antagonist of the IL-1R1 receptor) or anti-IL-1β monoclonal antibodies (reviewed in [10,24]); VX765 (an inhibitor of caspase 1, although its use has been dismissed because of its hepatic toxicity [16]); minocycline [25], or natural products such as epigallocatechin gallate (EGCG) or punicalagin ([26,27], reviewed in [28]), although caution should be taken when the possible utility of these compounds is considered, as, at least in the case of EGCG, it is regarded a pan-assay interfering compound [29]. In all these cases, the administration of any of these compounds decreases the appearance of seizures, at least in animal models of epilepsy (reviewed in [10,24]) (Table 1) (Figure 1, names in red).

(b) To reduce the HMGB1/TLR4 signaling pathway, some authors have used BoxA and P5779 compounds that are antagonists of the binding of HMGB1 to TLR4 or RAGE receptors; anti-HMGB1 monoclonal antibodies; natural products (glycyrrhizin, EGCG) to prevent the release of HMGB1; and a combination of N-acetyl-cysteine plus sulforaphane (an activator of the Nrf2 pathway) to decrease oxidative stress conditions, which would prevent the formation of the oxidized form of HMGB1 (disulfide form) that is the active one. These strategies have given good results in preventing the appearance of seizures in animal models of epilepsy (reviewed in [22,30,31]) (Table 1) (Figure 1).

### 2.2. Inflammasome

Specific DAMPs can be recognized by a specific type of cytosolic PRRs named nucleotide-binding oligomerization domain (NOD) and leucine-rich repeat receptors (NLRs). NLRs are classified in NODs, NLRPs, and IPAFs (NLRC4 plus NAIPs) [50,51,52]. In all the cases they are complexes consisting of a DAMP-receptor, an ASC (apoptosis-associated speck-like protein containing a C-terminal caspase recruitment domain (CARD)) adaptor and pro-caspase1. When they receive an activation signal, they release active caspase1 which processes the pro-forms of the inflammatory cytokines IL-1β and IL-18. The release of these cytokines is a two-step process: First, the activation of the NF-kB pathway (see above) activates the expression of pro-IL-1β and pro-IL-18. Then a second triggering event, which is normally either the activation of purinergic P2Xns or P2Yns by nucleotides (ATP, UTP, etc.), which opens Ca++ channels and allows the efflux of K+ from the cell, or the presence of reactive oxygen species (ROS) induces a change in the conformation of the NLRs receptors leading to an active form which converts inactive pro-caspase1 in its activated moiety [7,16,51,53] (Figure 2). The inflammasome plays a main role in microglia function, mainly through the activation of NLRP3, being microglia one of the main sources of IL-1β and IL-18 in the CNS [54,55,56].

Recently, a non-canonical inflammasome pathway has been described [15,57,58]. In this case, no NLRs are required, and the protein that plays a key role is caspase11 (in mice; caspase4/5 in humans). This caspase gets activated by a direct contact with LPS or still unknown endogenous factors. Upon activation, it releases an N-terminal proteolytic fragment from gasdermin D (gasdermin-Nterm), which is inserted in the plasma membrane producing pores. Through these pores, K+ exits the cells [15,58]. The decrease in intracellular K+ activates then NLRP3, producing IL-1β and IL-18 that are released from the cell throughout the gasdermin-N pores. Eventually, cells die by pyroptosis [15,52,59] (Figure 2).

Caspase11 expression is regulated by the TLR-TRIF-IRF3 pathway (see above), which produces the pro-inflammatory mediators type I-IFN and NF-kB which activate caspase11 expression [60]. Caspase11 has been identified in both astrocytes and microglia, and recent reports suggest a main role of the non-canonical pathway of the inflammasome in the appearance of seizures [61,62,63] (Figure 2).

### 2.3. The TNFα-TNFR1 Signaling Pathway

The TNFR1 (TNFα receptor 1) signaling pathway is induced by the binding of TNFα (tumor necrosis factor α) to this receptor. This binding induces the trimerization of the receptor and the recruitment of the TRADD (TNF-receptor type1-associated death domain protein) adaptor and the activation of the RIPK1 (receptor-interacting serine/threonine-protein kinase 1)–TRAF2 (TNFR-associated factor 2) complex which eventually will activate the TAK1/TAB(1,2,3) kinase complex leading to the activation of NF-kB and AP-1 (see above) [64] (Figure 3A).

The levels of TNFα are elevated in epileptic patients (reviewed in [61,62]), and probably contribute to their neuronal hyper-excitability because, on the one hand, TNFα increases the levels of glutamate in the brain, an excitatory neurotransmitter, since it reduces glutamate uptake by diminishing the levels of astrocytic glutamate transporter EAAT2 (excitatory amino acid transporter 2), increases the release of glutamate by glia by improving the function of the xCT cystine-glutamate antiporter, and increases the levels of astrocytic glutaminase, which produces glutamate from glutamine [65,66] (Figure 3B). On the other hand, TNFα increases the functionality of the glutamatergic AMPA (α-amino-3-hydroxy-5-methyl-4-isoxazolepropionic acid) and NMDA (N-methyl-D-aspartate) receptors in the post-synaptic neuron, leading to excitotoxicity. At the same time, TNFα induces the endocytosis of neuronal ionotropic GABAa (gamma-aminobutyric acid a) receptors, so the neurotransmission becomes more excitatory, leading to epilepsy [10,67,68].

In order to prevent the harmful action of TNFα, several authors have used anti-TNFα monoclonal antibodies (etanercept) and non-specific TNFα inhibitors such as dihydrothalidomide, nilotinib, and cannabinoids. All these treatments have proved to be successful in ameliorating seizures in animal models of epilepsy (reviewed in [32]) (Table 1).

### 2.4. The IL-6-Grp130/JAK-STAT Signaling Pathway

IL-6 triggers the activation of the Grp130/JAK (Janus kinase) receptor which induces the phosphorylation of STAT3 (*signal transducer and activator of transcription 3*), allowing its dimerization and its translocation to the nucleus where it activates the expression of pro-inflammatory mediators (e.g., SOCS3 (*suppressor of cytokine signaling 3*), cytokines, etc.). In addition, the Grp130/JAK kinase activates alternative pathways such as the MAPK (ERK, JNK, and p38) and the PI3K (phosphatidylinositol-3-kinase) pathways, the latter leading to the activation of the protein kinase B (AKT/PKB) which will inactivate the FOXO3 (forkhead box O3) transcription factor [69] (Figure 4A).

As in the case of TNFα, the activation of the IL-6 pathway leads to the accumulation of glutamate in the brain since it decreases astrocytic glutamate uptake by the EAAT2 transporters and promotes the astrocytic release of glutamate by improving the activity of the xCT cystine-glutamate antiporter. In addition, IL-6 increases the permeability of the blood–brain barrier (BBB), allowing the entrance of albumin and peripheral inflammatory cells to the brain parenchyma [7,10]. 

In order to prevent the deleterious effect of IL-6, several authors have proposed the use of an anti-IL-6 monoclonal antibody or the use of specific STAT3 inhibitors such as WP1066 [33,34] (Table 1).

### 2.5. The TGF-β-Receptor II Signaling Pathway

When the permeability of the BBB is affected, albumin enters into the brain parenchyma and triggers the activation of the TGF-β (transforming growth factor β) receptor I/II signaling pathway, which ends with the phosphorylation of the SMAD2/3 (small mothers against decapentaplegic 2/3) proteins, their dimerization, and translocation to the nucleus where they regulate the expression of a specific set of inflammatory-related genes. As a consequence, several astrocytic functions are affected (e.g., there is a decrease in the levels of Kir4.1 (inwardly rectifying K^+^ channel) and AQP4 (aquaporin 4), which affects the astrocytic buffering capacity to remove the K+ that is released as a consequence of neuronal transmission [3,5,70]; there is also a decrease in the astrocytic uptake of glutamate, etc.), which eventually produces a hyper-excitatory profile that promotes seizures (Figure 4B). In fact, alteration in the permeability of the BBB is one of the initial events in epilepsy [35,61,71,72]. A compromised BBB permeability also allows the entry of peripheral inflammatory leukocytes into the brain aggravating neuroinflammation [35].

To prevent the albumin-mediated activation of the TGF-β signaling pathway several authors have used losartan, an antagonist of the angiotensin receptor I, with positive results in the prevention of seizures [35] (Table 1).

### 2.6. The Chemokine Signaling Pathway

Chemokines are small proteins (around 8-14 kDa) that are produced upon conditions of inflammation. Some authors consider chemokines as the third major communication system in the brain, after neurotransmitters and neuropeptides [55,73]. They activate specific G-protein coupled receptors (GPCRs) which trigger the activation of MAPK (ERK, JNK, p38) and the activation of the PLCγ-PKC (phospholipase Cγ/protein kinase C) pathways, leading to the activation of the AP-1 and NF-kB routes, respectively. Chemokines also produce the inhibition of adenylate cyclase (AC), leading to a decrease in the levels of cAMP and an inhibition of the PKA (protein kinase A)-mediated pathway [73,74,75]. Some of the chemokine–receptor pairs have been related to epilepsy, being highly expressed in the hippocampus (reviewed in [17,18,61,62]) (Figure 5A). 

Some of examples of these are:

(a) CCL2-CCR2 (CCL2 receptor): They are expressed in astrocytes, microglia, and endothelial cells. As a consequence of the signaling pathways described above, there is an activation of microglia, which eventually releases IL-1β [76]. At the same time, the activation of the PLCγ-PKC pathway results in the release of Ca++ from intracellular stores, leading to cytotoxicity [77]. The expression of CCL2 has been reported in several forms of epilepsy [36], [37], and the levels of CCL2 in blood or cerebrospinal fluid (CSF) may be used as biomarkers of the progression of the disease [73]. Some compounds have been used in order to prevent CCL2 action: an antagonist of CCR2 (RS102895) and bindarit (a compound that decreases the expression of CCL2); they suppress LPS-induced seizures in animal models [36,37,38] (Table 1).

(b) CCL5/RANTES-CCR5: They are expressed mainly in astrocytes upon stimulation of microglia-derived TNFα and IL-1β. In this case, the signaling pathway goes mainly through the activation of the PLCγ-PKC pathway and the inhibition of AC, which produces a positive mobilization of intracellular Ca++ levels [32,74]. The expression of CCL5 has been reported in several forms of epilepsy, and the levels of CCL5 in blood or cerebrospinal fluid (CSF) could be used as biomarkers of the progression of the disease [74,78].

(c) CXCL10/IP10-CXCR3 (C-X-C Motif Chemokine Ligand 10/CXC receptor 3): They are expressed mainly in astrocytes upon stimulation with the combined action of IFNγ and TNFα. CXCL10/IP10 is a key player in astrocytic reactivity, acting in an autocrine and a paracrine mode, as CXCR3 is also expressed in neurons and microglia. The main result of the CXCL10-CXCR3 signaling pathway is an increase in the levels of intracellular Ca++ which leads to cytotoxicity. The expression of CXCL10 has been reported in several forms of epilepsy and the levels of CXCL10 in blood or cerebrospinal fluid (CSF) could be used as biomarkers of the progression of the disease [39,40,78,79,80]. Recently, a therapy based on the use of anti-CXCL10 monoclonal antibodies has shown some promises in the treatment of animal models of epilepsy [39,40,41] (Table 1).

### 2.7. The Lipocalin2-LCN2R Signaling Pathway

Lipocalin2 (Lcn2) is a protein of around 22 kDa which is produced mainly by astrocytes. It is considered as a strong marker of reactive astrocytes. Lcn2 activates the LCN2R receptor which is located in neurons, microglia, astrocytes, and epithelial cells. Thus, Lcn2 may act in an autocrine and a paracrine mode. In the brain, LCN2R is mainly expressed in the hippocampus where, when overactivated, it manifests its harmful effects [81,82]. Several inflammatory pathways convey in the activation of the expression of *LCN2*, mainly those mediated by NF-kB, MAPK, STATs, C/EBP (CCAAT-enhancer-binding proteins), and HIF1α (hypoxia-inducible factor 1-α) [76,83]. Binding of Lcn2 to LCN2R triggers the activation of the JAK-STAT3 and the IKK/NF-kB pathways (see above) leading to the expression of genes such as CXCL10, GFAP (glial fibrillary acidic protein, a component of the cytoskeleton), ITGB3 (integrin beta chain beta 3), etc. (Figure 5B). Activation of LCN2R in microglia leads to its activation, whereas in neurons promotes cytotoxicity [83].

High levels of expression of Lcn2 have been reported in several forms of epilepsy and the levels of Lcn2 in blood or cerebrospinal fluid (CSF) could be used as biomarkers of the progression of the disease [83,84].

## 3. Relationship between Reactive Astrocytes and Microglia in Neuroinflammation

Taking together all the information described above, it becomes clear that astrocytes and microglia participate actively in the process of neuroinflammation. At the moment, it is not possible to define an “order of events” in the neuroinflammatory process since astrocytes and microglia are so interconnected that the activity of one type of cell affects the activity of the other [76,85]. For example, astrocytes produce Lcn2 and CXCL10 that act respectively on microglial LCN2R and CXCR3 receptors to induce microglial activation [76]. They also produce HMGB1 that acts on TLR4 and RAGE receptors present in microglia and neurons. Astrocytes are also the main source of C3, a protein of the complement system which is used to target synapsis that needs to be removed, in the brain; C3 is recognized by the C3aR receptor present in microglia and this recruitment allows the pruning of the targeted synapses [17,18,76,86] (Figure 6).

In addition, reactive astrocytes enhance the production of specific markers such as glial fibrillary acidic protein (GFAP) and S100 proteins, a family of calcium-binding proteins, which are involved in the maintenance of cytoskeleton, although they also play a role in intracellular communication, regulation of cell cycle and energy metabolism. These proteins can be released to the extracellular media, reaching CSF and blood. In this sense, S100β levels have been proposed as a biomarker of neurological disorders, and it has been observed that the levels of S100β in serum increased in patients who experienced unfavorable seizure outcomes [87,88].

Microglia can also be activated by the release of ATP from dying neurons, which triggers activation of microglial purinergic receptors (P2Yns and P2Xns), or by the accumulation of glutamate or ROS in the brain [4,85]. Activated microglia induces then the expression of IL-1α, TNFα, and C1q (another component of the complement system), and they allow the conversion of resting astrocytes into type A1 reactive astrocytes [86,89,90,91]. Microglia also secretes cytokines, chemokines, and different types of interferons which affect the functionality of reactive astrocytes and neurons [92,93]. To prevent microglia-dependent activation of astrocytes, several authors have used pegylated exedin4 (NLY01), an agonist of glucagon-like peptide1 (GLP1) receptor, which reduces the levels of IL-1α, TNFα, and C1q [42,43]. In the same way, some reports suggest a beneficial effect of propranolol (an antagonist of β-adrenergic receptors) in preventing microglia activation [44] (Table 1).

Both astrocytes and microglia collaborate in synaptic pruning and phagocytosis of apoptotic cells. These processes are mediated by the presence of “find-me” signals of neuronal origin (e.g., ATP and other nucleotides, fractalkine, etc.) which attract and are recognized by microglial receptors (e.g., purinergic P2 × 7 receptor for ATP, CX3CR1 for fractalkine) [86,87,88,89,90,91,92,93,94]. Neuronal cells ready to be eliminated are also decorated on the external phase of their plasma membrane with “eat-me” signals such as C3, C1q, phosphatidylserine (PS), or ApoE, which are recognized by microglial CR3/Cd11b receptor (for complement proteins) and TREM2 (triggering receptor expressed on myeloid cells 2)/DAP12 (DNAX Adaptor Protein 12) and MerTk (MER tyrosine kinase) receptors (for PS). Binding of the “eat-me” signals to the corresponding receptors triggers different signaling cascades that eventually lead to the phagocytosis of the targeted cell [91,95,96].

In epileptic patients, a higher number of reactive glia (glial fibrillary acidic protein (GFAP+) astrocytes and ionized calcium-binding adapter molecule 1 (IBA1+) microglia) are found in the cornus ammonis 1 (CA1) and cornus ammonis 3 (CA3) layers of the hippocampus and the number of these cells correlates with higher neuronal death and greater severity of the symptoms [97,98,99] (Figure 6). Several authors indicate that microglia are the first type of cell to become activated by a seizure insult. Then, it propagates seizure-induced inflammatory responses and contributes to the pathogenesis of epilepsy [61,97,98,99,100]. In epilepsy, high levels of C1q, C3, and TREM2/DAP12 receptor have been found in hippocampal samples from different forms of human and animal models of epilepsy, suggesting an abnormal phagocytosis of synapses and/or neurons [91,95,96,101,102,103]. In order to prevent excessive phagocytosis, an anti-C1q monoclonal antibody has been obtained, and it has shown beneficial effects in animal models of epilepsy [45,46] (Table 1).

## 4. Neuroinflammation and Seizures

All the information described so far supports the notion that neuroinflammation and epilepsy are closely interconnected [11,17,72]. Seizures produce neuronal injury which releases ATP, glutamate, and reactive oxygen species (ROS). These are signals that are recognized by microglia and lead to its activation and eventually also to the activation of astrocytes, which ends with the combined production of pro-inflammatory mediators (IL-1β, IL-6, TNFα, chemokines, Lcn2, etc.). These compounds will initiate the specific inflammatory signaling cascades described above [33,67,104].

The release of pro-inflammatory mediators affects severely the functionality of astrocytes. The presence of IL-1β, TNFα, IL-6, etc., produces an increase in the levels of glutamate in the brain, due to a decrease in the functionality of the astrocytic glutamate transporters (by reducing glutamate uptake), and an increase in the release of glutamate from the astrocytes and microglia, due to enhanced formation of intracellular astrocytic glutamate because of increased glutaminase activity and an increase in the activity of the astrocytic xCT cystine-glutamate antiporter [5,104,105,106]. At the post-synaptic neuronal level, these pro-inflammatory mediators improve the functionality of the glutamate AMPA and NMDA receptors [68,104,107]. The presence of high levels of glutamate and the enhanced activity of the glutamate receptors at the post-synaptic neuron leads to excitotoxicity and hyper-excitability. In addition, these pro-inflammatory mediators also diminish the presence of neuronal GABAa receptors at the plasma membrane, leading to a decrease in the functionality of GABAergic inhibitory neurons. This combined action improves excitatory transmission and reduces inhibitory firing, resulting in enhanced susceptibility to seizures. Moreover, the pro-inflammatory mediators also affect other astrocytic functions. They decrease the expression of Kir4.1 and AQP4 transporters, decreasing in this way the functionality of the astrocytes to buffer the excess of K+ that is released after neuronal firing [5,104,108]. They also decrease the activity of the glutamine synthase, preventing the production of glutamine from glutamate and reducing in this way the rate of the glutamate–glutamine cycle. This eventually leads to lower production of GABA neurotransmitter in the GABAergic neurons, diminishing its firing [70,109]. Finally, the pro-inflammatory mediators severely affect the permeability of the BBB; this allows the entry of albumin, and peripheral inflammatory leukocytes which will aggravate inflammation. In summary, the presence of pro-inflammatory mediators forces a hyper-excitatory state that will end with the appearance of seizures, closing in this way this positive vicious feedback cycle (Figure 6).

## 5. Therapeutic Approaches to Ameliorate Seizures Based on Glia Functionality

Glia may contribute to seizures in three main scenarios: glutamate homeostasis, oxidative stress, and production of pro-inflammatory mediators [5,47]. Different therapeutic opportunities can be envisioned in each of these conditions. In the case of glutamate homeostasis, different compounds have been described in the literature which affect distinct aspects of glutamate turnover: compounds that improve the expression of the glutamate transporters (e.g., ceftriaxone) [47], that improve their activity (e.g., riluzole) [48], or that prevent the function of the xCT cystine-glutamate antiporter (e.g., sulfasalazine) decreasing in this way the astrocytic release of glutamate [5,48,49] could be useful to control the levels of glutamate in the brain (Table 1).

In the case of oxidative stress, the administration of compounds that decrease the levels of ROS (e.g., resveratrol) or the combined action of N-acetyl-cysteine plus sulforaphane (an activator of the Nrf2 pathway) could help in decreasing the conditions of oxidative stress [3,20,48] (Table 1), although caution should be taken when the possible utility of resveratrol is considered, as it is regarded as a pan-assay interfering compound ([110]; https://blogs.sciencemag.org/pipeline/archives/2015/11/11/screen-carefully).

Finally, as neuroinflammation plays a major role in inducing seizures, the administration of different compounds that affect the specific inflammatory signaling cascades which are activated in a particular form of epilepsy could control the production of pro-inflammatory mediators (see above) (Table 1). Perhaps, the combined action of therapies targeting different key pro-inflammatory molecules could provide a major suppression of inflammatory networks and may lead to new promising therapeutic avenues to prevent the appearance of seizures in those patients that are resistant to the conventional treatment with ASDs [10,106,111].

## Figures and Tables

**Figure 1 ijms-21-04096-f001:**
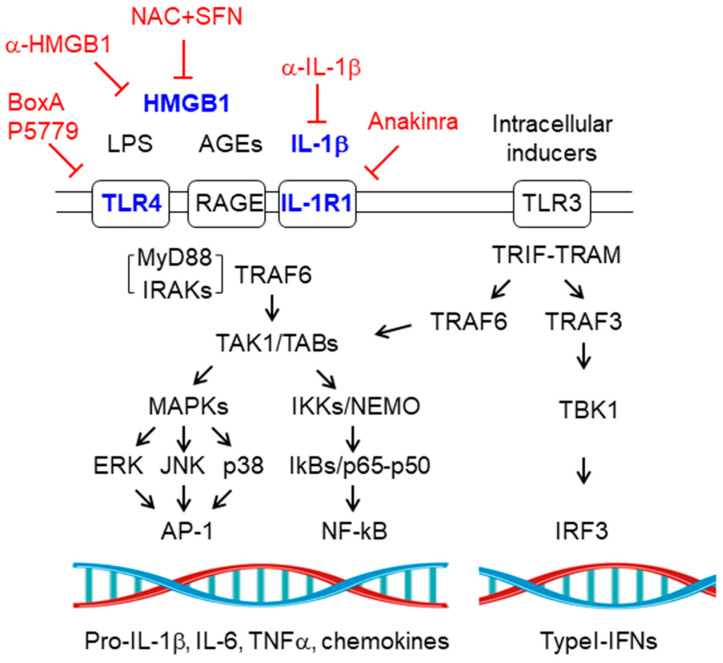
Toll-like receptors (TLRs) signaling pathway. Schematic view of the components of the TLRs signaling pathways. See text for details. TLR4, RAGE, and IL-1R1 are mainly located at the plasma membrane, whereas TLR3 is located in intracellular vesicles. Components of the pathway that are increased in epilepsy are in blue. Therapeutic strategies to prevent the pathway are in red. NAC+SFN: N-acetyl-cysteine plus sulforaphane; α-HMGB1: anti-high mobility group box 1 monoclonal antibody; α-IL-1β: anti-interleukin-1β monoclonal antibody. Pointed arrows indicate direction of the signaling pathway. Blunt arrows indicate inhibition.

**Figure 2 ijms-21-04096-f002:**
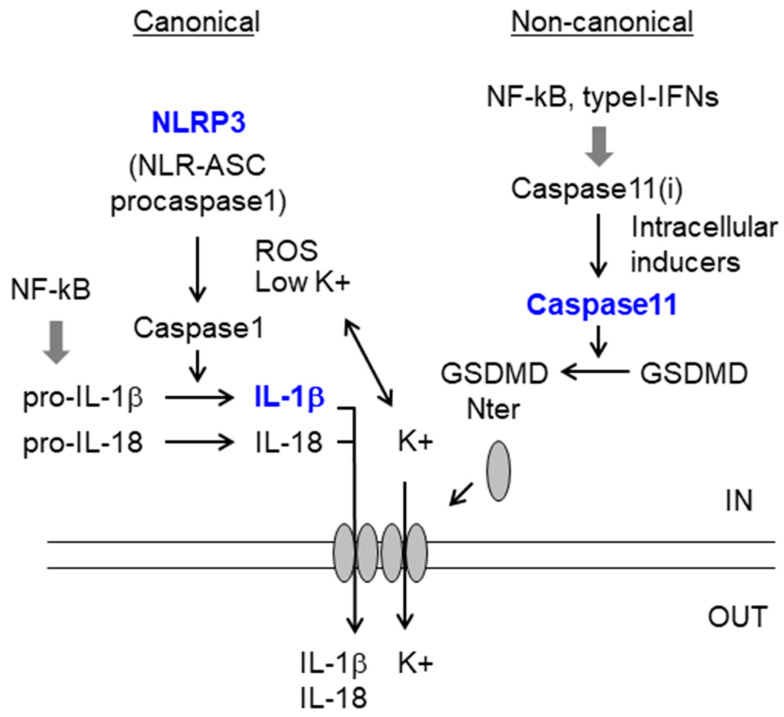
Canonical and non-canonical inflammasome pathways. The canonical pathway is a two-step process: first, the NF-kB pathway activates the expression of pro-IL-1β and pro-IL-18; then activation of the inflammasome (e.g., NLRP3) is required (e.g., by K+ efflux or by ROS) to activate caspase1. In the non-canonical pathway, the activation of caspase11 by intracellular inducers leads to the generation of Gasdermin-Nterm (GSDMD-Nterm), which translocates to the membrane and creates pores that allow the efflux of K+ and the release of mature cytokines. See text for details. Components of the pathway that are increased in epilepsy are in blue. Caspase 11 (i), inactive form. Pointed arrows define the direction of the signaling pathways. Wide grey arrow indicate transcriptional regulation.

**Figure 3 ijms-21-04096-f003:**
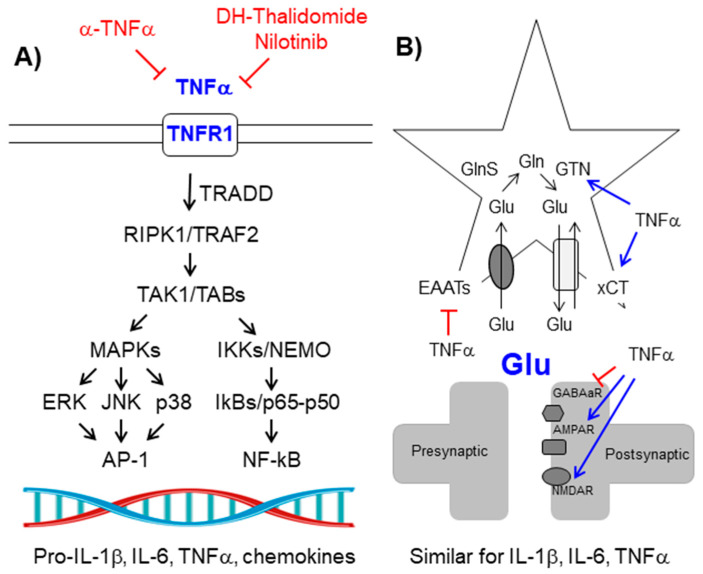
Tumor necrosis factor α (TNFα)/TNFR1 signaling pathway. (**A**) Schematic view of the components of the signaling pathway. See text for details. Components of the pathway that are increased in epilepsy are in blue. Therapeutic strategies to prevent the pathway are in red. (**B**) TNFα enhances neuronal glutamatergic transmission: On the one hand, it increases the levels of glutamate (Glu), and on the other hand, it enhances the activity of postsynaptic glutamate receptors. See the text for details. Pointed arrows indicate direction of the signaling pathway. Blunt arrows indicate inhibition.

**Figure 4 ijms-21-04096-f004:**
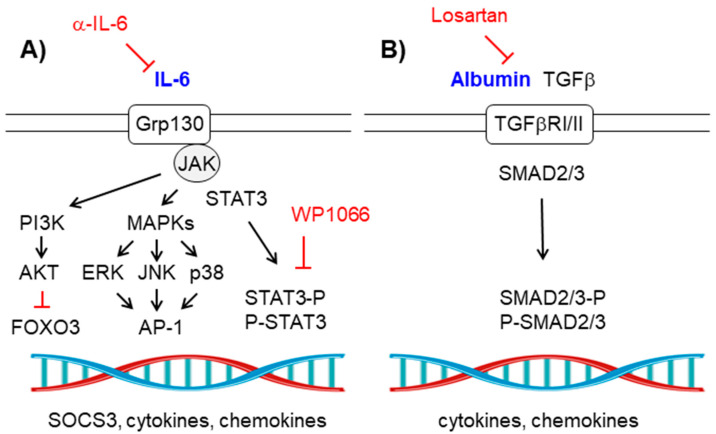
(**A**) Interleukin (IL)-6/Grp130-JAK-STAT signaling pathway. Schematic view of the components of the signaling pathway. View text for details. Components of the pathway that are increased in epilepsy are in blue. Therapeutic strategies to prevent the pathway are in red. (**B**) Albumin-TGFβ/TGFβ-RI/II signaling pathway. Schematic view of the components of the pathway. See text for details. Components of the pathway that are increased in epilepsy are in blue. Therapeutic strategies to prevent the pathway are in red. Pointed arrows indicate direction of the signaling pathway. Blunt arrows indicate inhibition.

**Figure 5 ijms-21-04096-f005:**
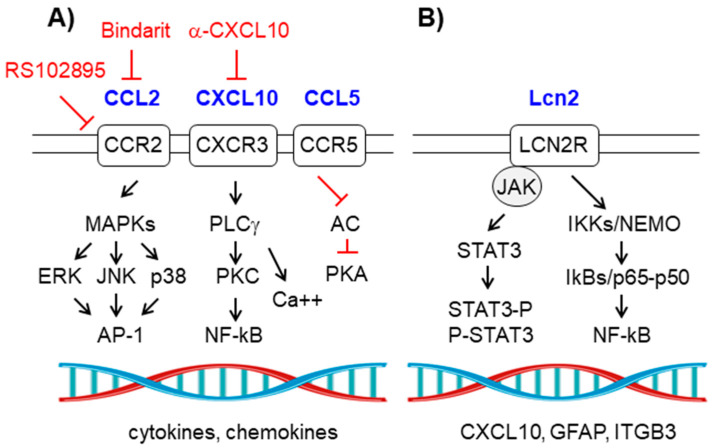
(**A**) Chemokine signaling pathway. CCR2, CCR5, and CXCR3 belong to the group of G-protein coupled receptors (GPCRs). Schematic view of the components of the signaling pathway; see text for details. Components of the pathway that are increased in epilepsy are in blue. Therapeutic strategies to prevent the pathway are in red. (**B**) Lcn2/LCN2R signaling pathway. Schematic view of the components of the pathway. See text for details. Components of the pathway that are increased in epilepsy are in blue. Pointed arrows indicate direction of the signaling pathway. Blunt arrows indicate inhibition.

**Figure 6 ijms-21-04096-f006:**
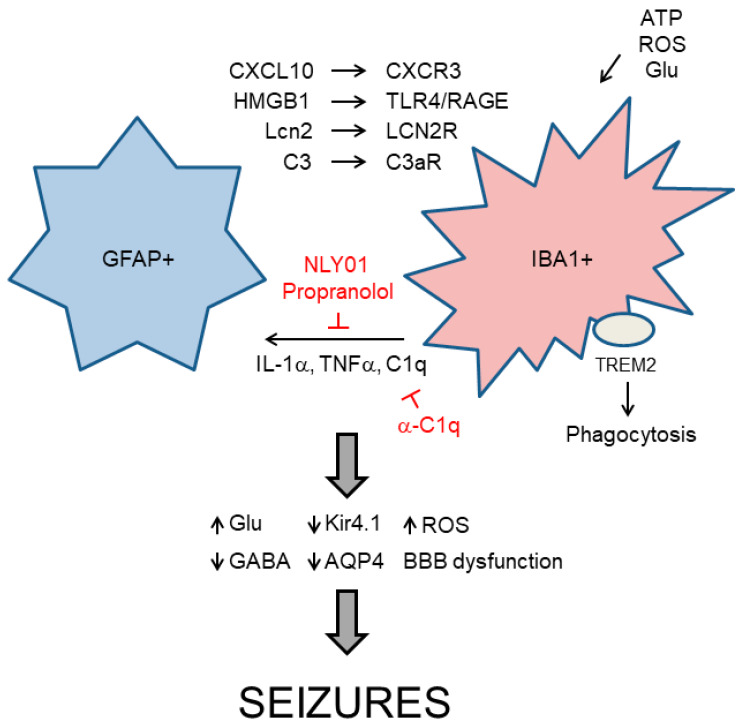
Functional relationship between GFAP+ (glial fibrillary acidic protein) reactive astrocytes and IBA+ (ionized calcium-binding adapter molecule) activated microglia. Reactive astrocytes produce critical components that activate microglia (e.g., CXCL10, HMGB1, Lcn2, C3, etc.). These components are recognized by specific microglial receptors (CXCR3, TLR4/RAGE, LCN2R, C3aR, respectively) that trigger signaling cascades leading to microglia activation. Microglia can also get activated by increases in the levels of ATP, reactive oxygen species (ROS), or glutamate in the medium. Activated microglia produces IL-1α, TNFα, and C1q, which transform resting astrocytes into type A1 reactive astrocytes. Therapeutic strategies to prevent this action are in red. In addition, activated microglia participates in the phagocytosis of neurons using specific receptors (e.g., TREM2—triggering receptor expressed on myeloid cells 2) that recognize specific “eat-me” signals. As a consequence of astrocyte and microglia reactivity, critical mediators in glial physiology are altered, leading to seizures. Pointed arrows indicate direction of the signaling pathway. Blunt arrows indicate inhibition.

**Table 1 ijms-21-04096-t001:** Therapeutic opportunities based on ameliorating reactive glia-derived neuroinflammation.

Signaling Pathway	Compound	Mechanism of Action	Reference
IL-1β/IL-1R1	Anakinra	Antagonist of IL-1R1 receptor	Reviewed in [10,24]
	Anti-IL-1β Mabs	Blocking IL-1β	Reviewed in [10,24]
	Minocycline	unknown	[25]
	EGCG, punicalagin	unknown	[26,27,28]
HMGB1/TLR4-RAGE	BoxA, P5779	Antagonist of TLR4-RAGEs	Reviewed in [22,30,31]
	Anti-HMGB1 Mabs	Blocking HMGB1	Reviewed in [22,30,31]
	Glycyrrhizin, EGCG	unknown	Reviewed in [22,30,31]
	N-acetyl-cysteine + sulforaphane	Prevents the formation of active disulfide form of HMGB1	Reviewed in [22,30,31]
TNFα/TNFR1	Anti-TNFα Mabs	Blocking TNFα	Reviewed in [32]
	Dihydrothalidomide, Nilotinib, Cannabinoids	Non-specific TNFα inhibitors	Reviewed in [32]
IL-6/Grp130-JAK-STAT	Anti-IL-6 Mabs	Blocking IL-6	[33,34]
	WP1066	STAT3 inhibitor	[33,34]
Albumin/TGF-β RI/II	Losartan	Blocking TGF-β signaling	[35]
CCL2/CCR2	RS102895	Antagonist of CCR2	[36,37,38]
	Bindarit	Decreases the expression of CCL2	[36,37,38]
CXCL10/CXCR3	Anti-CXCL10	Blocking CXCL10	[39,40,41]
Microglia activation	NLY01	Prevents microglial production of IL1α, TNFα, C1q	[42,43]
	Propranolol	Prevents microglia activation	[44]
	Anti-C1q Mabs	Blocks C1q	[45,46]
Controlling the levels of glutamate	Ceftriaxone	Enhances EAATs expression	[47]
	Riluzole	Improves EAATs activity	[48]
	Sulfasalazine	Decreases activity of the xCT antiporter	[5,48,49]
Controlling the levels of ROS	Resveratrol	antioxidant	[3,20,48]
	N-acetyl-cysteine + sulforaphane	antioxidant	[3,20,48]

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
