# Peer review of "Reactive Glia Inflammatory Signaling Pathways and Epilepsy"

_ijms, 2020, doi:10.3390/ijms21114096_

Round 1

Reviewer 1 Report

The manuscript by Pascual Sanz and Maria Adelaida Garcia-Gimeno, titled, ‘Reactive glia Inflammatory signaling pathways and 2 epilepsy’ is written relatively well. The authors have reviewed the many different signalling cascades related to glial cells and explained their roles underlying inflammation in epilepsy.

I have a few minor comments that I hope will help the authors improve their review.

  1. As the list of references in this manuscript shows, many other reviews have been published in this field. I would like the authors to justify writing and publishing of this review. I would like the authors to highlight the features in this review that may not have been covered in, or offered by, other similar review articles.
  2. In agreement with the comment above, I would like the authors to cite primary research articles instead of reviews in their manuscript. For example, references 10, 24, 28, 22, 29, 30, to name a few—cited in Table 1 and in the text—do not seem to be primary research articles, but reviews.
  3. Following on from comment 2, I would like the authors to identify the background of certain evidence reviewed in their article. For example, when presenting experimental findings and facts, I would like the authors to specify whether the evidence was obtained using cell-culture models, animal models, or human experimentation. This could be appreciated by the readers.
  4. Evidence suggests that EGCG is a pan-assay interfering compound (PAIN). I would encourage the authors to be critical of the literature when presenting data on PAINS. See, ‘Kanlaya R, Thongboonkerd V. Molecular Mechanisms of Epigallocatechin-3-Gallate for Prevention of Chronic Kidney Disease and Renal Fibrosis: Preclinical Evidence. Curr Dev Nutr. 2019 Aug 29;3(9):nzz101. doi: 10.1093/cdn/nzz101. PMID: 31555758; PMCID: PMC6752729.’
  5. The same as comment 4, resveratrol is also a PAINS (https://blogs.sciencemag.org/pipeline/archives/2015/11/11/screen-carefully; and Helgi I. Ingólfsson, Pratima Thakur, Karl F. Herold, E. Ashley Hobart, Nicole B. Ramsey, Xavier Periole, Djurre H. de Jong, Martijn Zwama, Duygu Yilmaz, Katherine Hall, Thorsten Maretzky, Hugh C. Hemmings, Carl Blobel, Siewert J. Marrink, Armağan Koçer, Jon T. Sack, and Olaf S. Andersen ACS Chemical Biology 2014 9 (8), 1788-1798 DOI: 10.1021/cb500086e). Please avoid PAINS as evidence. These chemicals should be doubted instead of praised.
  6. I would like the authors to discuss the role of the inflammation-associated S100 protein as an example of DAMPs in epilepsy. Evidence exists for S100B, and potentially for S100A8, S100A9, and S100A12.
  7. In-text citations and references do not seem to have the correct style.
  8. Some punctuation inconsistency exists in the manuscript; for example, underscore dash has been used instead of en-dash. Sentence structure can be improved, and overall, the English in the manuscript will benefit from proper editing.

Author Response

Reviewer 1:

The manuscript by Pascual Sanz and Maria Adelaida Garcia-Gimeno, titled, ‘Reactive glia Inflammatory signaling pathways and epilepsy’ is written relatively well. The authors have reviewed the many different signalling cascades related to glial cells and explained their roles underlying inflammation in epilepsy. I have a few minor comments that I hope will help the authors improve their review.

1.- As the list of references in this manuscript shows, many other reviews have been published in this field. I would like the authors to justify writing and publishing of this review. I would like the authors to highlight the features in this review that may not have been covered in, or offered by, other similar review articles.

            There are indeed excellent reviews on the field that have been published recently. However, in our work we stress the role that glia has on the development of epilepsy. In addition, we consolidate the relationship between neuroinflammation and epilepsy and define the main cell signaling pathways that operate in glial cells. This review puts together all this information to define possible therapeutic opportunities to prevent epileptic dysfunction.

            As these notions were already indicated at the end of the Introduction section, in our opinion, including them again in the text would be redundant.

2.- In agreement with the comment above, I would like the authors to cite primary research articles instead of reviews in their manuscript. For example, references 10, 24, 28, 22, 29, 30, to name a few—cited in Table 1 and in the text—do not seem to be primary research articles, but reviews.

            The reviewer is right. In some cases, we have used a review on a topic instead of the original reference. The reason for that is to reduce the number of references in a manuscript that deals with a huge number of cell signaling pathways and their associated repercussions in cell physiology. In the revised version we have indicated “reviewed in [Ref]” to indicate this fact.

            These modifications can be found throughout the manuscript.

3.- Following on from comment 2, I would like the authors to identify the background of certain evidence reviewed in their article. For example, when presenting experimental findings and facts, I would like the authors to specify whether the evidence was obtained using cell-culture models, animal models, or human experimentation. This could be appreciated by the readers.

            Following the reviewer’s suggestion, we have defined the model used to support the mentioned evidence.

            These modifications can be found in lanes 139-142, 144, 347-348, 426-427.

4.- Evidence suggests that EGCG is a pan-assay interfering compound (PAIN). I would encourage the authors to be critical of the literature when presenting data on PAINS. See, ‘Kanlaya R, Thongboonkerd V. Molecular Mechanisms of Epigallocatechin-3-Gallate for Prevention of Chronic Kidney Disease and Renal Fibrosis: Preclinical Evidence. Curr Dev Nutr. 2019 Aug 29;3(9):nzz101. doi: 10.1093/cdn/nzz101. PMID: 31555758; PMCID: PMC6752729.

            We want to thank the reviewer very much for his/her comment. We have added a sentence indicating that EGCG is considered a pan-assay interfering compound. It says: “although caution should be taken when the possible utility of these compounds is considered as, at least in the case of EGCG, it is regarded as a pan-assay interfering compound [29]”.

This modification can be found in lanes 151-153.

5.- The same as comment 4, resveratrol is also a PAINS (https://blogs.sciencemag.org/pipeline/archives/2015/11/11/screen-carefully; and Helgi I. Ingólfsson, Pratima Thakur, Karl F. Herold, E. Ashley Hobart, Nicole B. Ramsey, Xavier Periole, Djurre H. de Jong, Martijn Zwama, Duygu Yilmaz, Katherine Hall, Thorsten Maretzky, Hugh C. Hemmings, Carl Blobel, Siewert J. Marrink, Armağan Koçer, Jon T. Sack, and Olaf S. Andersen ACS Chemical Biology 2014 9 (8), 1788-1798 DOI: 10.1021/cb500086e). Please avoid PAINS as evidence. These chemicals should be doubted instead of praised.

Similarly, as above, we have added a sentence indicating that resveratrol is considered a pan-assay interfering compound. It says: “although caution should be taken when the possible utility of resveratrol is considered, as it is regarded as a pan-assay interfering compound ([110], https://blogs.sciencemag.org/pipeline/archives/2015/11/11/screen-carefully)”.

            This modification can be found in lanes 478-480.

6.- I would like the authors to discuss the role of the inflammation-associated S100 protein as an example of DAMPs in epilepsy. Evidence exists for S100B, and potentially for S100A8, S100A9, and S100A12

     We want to thank the reviewer for raising this point. We have discussed the role of S100 proteins in the following way: “In addition, reactive astrocytes enhance the production of specific markers such as glial fibrillary acidic protein (GFAP) and S100 proteins, a family of calcium-binding proteins, which are involved in the maintenance of cytoskeleton, although they also play a role in intracellular communication, regulation of cell cycle and energy metabolism. These proteins can be released to the extracellular media, reaching CSF and blood. In this sense, S100b levels have been proposed as a biomarker of neurological disorders, and it has been observed that the levels of S100b in serum increased in patients who experienced unfavorable seizure outcomes [87-88]”.

This modification can be found in lanes 391-397.

7.- In-text citations and references do not seem to have the correct style.

We have corrected the in-text citations according to the Instructions for Authors of the IJMS.

These modifications can be found throughout the manuscript.

8.- Some punctuation inconsistency exists in the manuscript; for example, underscore dash has been used instead of en-dash. Sentence structure can be improved, and overall, the English in the manuscript will benefit from proper editing.

We have amended all typographical errors and punctuation inconsistencies we found in the text. The revised manuscript has been revised by a native English service.

The modifications related to dash can be found in lanes 263, 312 and 350. The rest of the punctuation modifications can be found throughout the text.

Reviewer 2 Report

The review by Pascual Sanz and Maria Adelaida Garcia-Gimeno, reviewed the main glial signaling pathways involved in neuroinflammation, how they are affected in epileptic conditions and the therapeutic opportunities they offer to prevent these disorders.

The review is clearly written, its original and of interest in its field.

I recommend that the article be accepted with minor revision.

  • The authors should better check the manuscript for any typographical errors;
  • Please remove the same sentence such as “Neuroinflammation and epilepsy are interconnected” and/or “In this work, we review the main glial signaling pathways involved in neuroinflammation, how they are affected in epileptic conditions and the therapeutic opportunities they offer to prevent these disorders.”
  • The authors talks about inflammatory pathway MyD88-dependent pathway but they don’t talk about the role of this pathway during epilepsy
  • Same for the inflammosome

Author Response

Reviewer 2:

The review by Pascual Sanz and Maria Adelaida Garcia-Gimeno, reviewed the main glial signaling pathways involved in neuroinflammation, how they are affected in epileptic conditions and the therapeutic opportunities they offer to prevent these disorders. The review is clearly written, it is original and of interest in its field. I recommend that the article be accepted with minor revision:

1.- The authors should better check the manuscript for any typographical errors.

            We have amended all typographical errors we have found.

2.- Please remove the same sentence such as “Neuroinflammation and epilepsy are interconnected” and/or “In this work, we review the main glial signaling pathways involved in neuroinflammation, how they are affected in epileptic conditions and the therapeutic opportunities they offer to prevent these disorders”.

            Following the reviewer’s suggestion, we have deleted the sentence “Neuroinflammation and epilepsy are interconnected”.

This modification can be found in lane 56.

3.- The authors talks about inflammatory pathway MyD88-dependent pathway but they don’t talk about the role of this pathway during epilepsy.

            In the point 2.1.1 of the manuscript, we describe the MyoD88-dependent pathway of the Toll-like receptor signaling pathway. As a final result of its activation, both NF-kB and AP1 transcriptional factors are activated, which results in the expression of pro-inflammatory mediators. Then, at the end of the section, we indicate: “It has been described that activation of the TLR4 and IL-1R1 receptors plays a key role in epileptogenesis. In fact, patients suffering from mesial temporal lobe epilepsy (MTLE) present higher levels of TLR4, IL-1R1, IL-1b, and HMGB1 in their hippocampus (reviewed in [17]) (Figure 1, names in blue). Changes in the levels of HMGB1 in the brain are mirrored in blood, so measuring blood levels of HMGB1 in patients can predict with high accuracy the risk of developing epilepsy (reviewed in [22]). In the same way, changes in the levels of IL-1b in blood or cerebrospinal fluid (CSF) in human samples could also be used as biomarkers of the progression of the disease (reviewed in [23])”. Lanes 138-145.

Therefore, with all our respects we consider that we have covered the importance of the MyD88 pathway in epilepsy.

4.- Same for the inflammosome.

            In our opinion, we have described in the manuscript the connection between inflammasome and epilepsy. In Fig. 2 we indicated that components of the inflammasome such as NLRP3, IL-1b , and caspase 11 that are increased in epilepsy, and referred to the text for more information. Lanes 218-230.

Therefore, with all our respects we consider that we have covered the importance of the inflammasome pathway in epilepsy.